# Electromechanical Performance of Biocompatible Piezoelectric Thin-Films

**S. Ranjan Mishra, Soran Hassani Fard, Taha Sheikh and Kamran Behdinan \***

Advanced Research Laboratory for Multifunctional Lightweight Structures (ARL-MLS),
Department of Mechanical and Industrial Engineering, University of Toronto (U of T),
Toronto, ON M5S 3G8, Canada; sr.mishra@mail.utoronto.ca (S.R.M.); s.hassanifard@mail.utoronto.ca (S.H.F.);
t.sheikh@mail.utoronto.ca (T.S.)
**\*** Correspondence: kamran.behdinan@utoronto.ca

**Abstract:** The present study analyzed a computational model to evaluate the electromechanical properties of the AlN, BaTiO$_3$, ZnO, PVDF, and KNN-NTK thin-films. With the rise in sustainable energy options for health monitoring devices and smart wearable sensors, developers need a scale to compare the popular biocompatible piezoelectric materials. Cantilever-based energy harvesting technologies are seldom used in sophisticated and efficient biosensors. Such approaches only study transverse sensor loading and are confined to fewer excitation models than real-world applications. The present research analyses transverse vibratory and axial-loading responses to help design such sensors. A thin-film strip (50 × 20 × 0.1 mm) of each sample was examined under volumetric body load stimulation and time-based axial displacement in both the d$_{31}$ and d$_{33}$ piezoelectric energy generation modes. By collecting evidence from the literature of the material performance, properties, and performing a validated finite element study to evaluate these performances, the study compared them with lead-based non-biocompatible materials such as PZT and PMN-PT under comparable boundary conditions. Based on the present study, biocompatible materials are swiftly catching up to their predecessors. However, there is still a significant voltage and power output performance disparity that may be difficult to close based on the method of excitation (i.e., transverse, axial, or shear. According to this study, BaTiO$_3$ and PVDF are recommended for cantilever-based energy harvester setups and axially-loaded configurations.

**Keywords:** biomaterial; energy harvesting; piezoelectric effect; sustainable energy; wearable sensors



## 1. Introduction

Implantable devices are of utmost important to increase the patient's lifespan in the form of artificial remedy tools, implantable cardioverter defibrillators (ICDs), and cardiac pacemakers as bioelectronics [1–4]. Such solutions warrant the usage of biocompatible materials for implantable in vivo–vitro applications [5]. This study utilized extensively applied piezoelectric materials such as AlN, BaTiO$_3$, ZnO, PVDF, and KNN-NTK [6–8]. The inception of bioelectronics was led by battery-powered devices and has been developed to improve the device performances. For instance, the storage capacity enhancement and reduction in the size of batteries has increased the lifespan of pacemakers up to 7–10 years, but still demands surgical replacement after this time frame [9,10]. These surgeries expose the patient to infection, pain and discomfort, and financial problems. Implantable medical devices demand high-energy density, small dimensions, longer lifespan, and biocompatibility. Possible sources considered by researchers are biofuel cells. These generate electricity by using renewable organic sources such as amalyum or glucose as biocatalysts [11,12]. Initially, these fuel cells have only used redox mediators, producing only a small amount of energy, which was later increased due to many new biofuel cells such as

alkaline fuel cells (AFCs), solid oxide fuel cells (SOFCs), and microbial fuel cells (MFCs) [13–15]. However, the demand for very small and thin devices that are flexible to accommodate the complex contour of the human body and utilizes free energy has led many researchers to investigate piezoelectric thin-film based EHs. To answer these challenges, researchers have been studying the alternatives to battery powered devices such as energy harvesting (EH) devices. These devices are capable of utilizing biomechanical movements such as muscle motions, lungs motion, or blood circulation to convert into electrical power output [16,17]. The piezoelectric effect is one of the techniques utilized to convert the mechanical energy into electrical power at a particular frequency range [18,19]. However, such EHs have limited applications because of their bulkiness, which restricts their contact with furrowed and undulated body surfaces. Flexoelectric devices utilize the polarization due to the point wise variation of strain throughout the entire domain of the material [20]. The flexoelectric coupling is between the polarization and strain gradient in lieu of a homogeneous strain; because of this difference, flexoelectricity is more advantageous than piezoelectricity. However, the flexoelectric effect is more prominent in materials having high wave acoustic transmission [21]. Surface wave propagation methods such as Love waves, shear horizontal wave, or Rayleigh wave have been dominantly used to investigate the properties and behavior of anisotropic, orthotropic, or crystals [22,23]. The field is still under investigation to manufacture sensitive sensors for biomedical applications that are potentially better than piezoelectric EHs. Piezoelectric films could be considered as thin-films that can be accommodated in the development of nanoscale or microscale devices. These thin-films are also characterized by their ability to be micromachined and incorporated into micromechanical systems (MEMS) and nanoelectromechanical systems (NEMS) [5,24]. Therefore, it can be said that the thickness of the thin-films lies at the micro- and nanoscales (100 nm to 390 um) [25,26]. Thus, such a small thickness decreases the bulk piezoelectric properties of these films, requiring novel materials with better piezoelectric coefficients together with the sensitivity of the devices [27,28]. Therefore, thin-film and flexible EHs are a necessity for sophisticated biomedical applications.

Piezoelectric energy harvesting (PEH) is one of the efficient techniques that is being used for powering low-power mobile/portable electronics, wearable, and implantable devices. The conventional electrochemical batteries and micro-fuel cells failed to satisfy the need for power due to their limited lifespan and periodic charging and replacement. The PEHs offer the opportunity to convert mechanical energy into electrical power output. There are several materials being used as piezoelectric parts for the purpose of harvesting energy in PEH devices. The $Pb(Zr,Ti)O_3$ (PZT) is the most widely-used piezo-ceramic material, which is an inorganic compound of $PbZrO_3$ (PZ) and $PbTiO_3$ (PT). It has excellent piezoelectric properties and is readily commercially available in comparison to other ceramic-based piezoelectric materials such as aluminum nitride (AlN) or zinc oxide (ZnO). Fraga et al. compared the piezoelectric properties of a wide range of piezo-ceramic and semiconductor materials [29]. The authors reported that while PZT possessed the highest piezoelectric coefficients, AlN and ZnO were more suitable for undergoing extreme loading/environmental conditions. Researchers have also employed PZT piezo-ceramics in thin- or thick-films depending on the targeted applications in energy harvester devices [30–33].

To improve the PVDF piezoelectric properties, there is significant research being conducted by incorporating different materials with PVDF such as reduced graphene oxide (rGO), TrFE. The author investigated the effect of reduced graphene oxide (rGO) on the piezoelectric properties of PVDF. The PVDF thin-film was prepared by soft lithography, followed by low temperature phase inversion. It is reported that the piezoelectric response of the PVDF was increased from 64 pm/V to 87 pm/V for the rGO loaded PVDF [34]. One study reported that the nanofillers at low concentration amalgamated with PVDF enhanced the $d_{33}$ coefficient. Such nanofillers were reported as carbon nanotubes (CNTs), graphene, metal-oxide based nanofillers, which demonstrate the enhanced piezoelectricity due to the nucleation of the polar piezoelectric $\beta$ and $\gamma$ phases [26,35,36]. There are

many other important studies focused on the PVDF-TrFE piezoelectric applications in biomedical sensing and generators. The important reason in the improvement of such properties is due to the variation in the electroactive phase content caused by nano- and microstructuring of the PVDF-TrFE [35–37].

There has been an extensive number of research reported for harvesting energy for biomedical applications. Pillatsch et al. studied a scalable piezoelectric impulse-excited energy harvester for human body excitation [38]. The authors discussed the principle of impulse excitation and introduced a centimeter-scale functional model as a proof-of-concept. The study also tested the voltage regulation at a frequency of 2 Hz and observed the maximum power output of 2.1 mW at an acceleration of 2.7 m/s$^2$. Halim et al. investigated a non-resonant, frequency up-converted electromagnetic energy harvester from human-body-induced vibration for hand-held smart system applications [39]. The study proposed a design capable of harvesting significant power from human body-induced vibration such as hand-shaking. It was observed that the device produced 110 µW average power with 15.4 µW cm$^{-3}$ average power density, from low frequency (<5 Hz) vibrations.

AlN piezo-ceramics have received increased attention in power generation for the device in microelectromechanical systems in the forms of thin/thick films with different structures [40–42]. Cao et al. used deposited AlN thin-films on stainless steel substrates using an electron cyclotron resonance technique, and fabricated millimeter-scale piezoelectric generators [43]. The authors reported the output power and resonant frequency of the devices to be 5.130 µW and 69.8 Hz, respectively, when vibrated at 1 g acceleration connected with the 0.7 MΩ electric loading. Sharma et al. studied a micro-electro-mechanical system (MEMS) vibration energy harvester using a AlN on a silicon wafer sandwiched between the top and bottom electrodes [44]. The choice of the piezoelectric materials depends on the application. However, the PZT-based PEHs had the biggest disadvantage of containing toxic materials, therefore limiting their application in biomedical industries.

Pillatsch et al. studied a wireless power transfer system for a human motion energy harvester and presented a mechanism with a rotational piezoelectric energy harvester wirelessly via a magnetic reluctance coupling to an external driving rotor with one or more permanent magnet stacks attached [45]. The authors observed that the optimal power output into a resistive load was over 100 µW at a frequency of 25 Hz. Fan et al. studied harvesting energy from ultra-low frequency vibrations and human motion through a monostable electromagnetic energy harvester [46]. They conducted experiments on a treadmill and showed that the fabricated prototype could generate approximately 0.5 mW of power under walking and 0.7 mW of power under running. Gao et al. developed a bimorph piezoelectric bending beam-based energy harvester to scavenge energy from the human knee motion for powering body-worn electronics such as smartwatches and health monitors [47]. The study showed that the average power could reach 12.79 mW for level-ground walking, 9.91 mW for stair descending, and 7.70 mW for stair ascending. There is somehow a direct relationship between normalized power density and device volume for some piezoelectric materials such as PZT, AlN, and PVDF [48]. This means that by changing the device volume when other factors remain the same, the power output changed accordingly. Another study investigated the effect of the ZnO films on the output voltage of flexible ZnO/Kapton/Al/ZnO/Al structures. The ZnO films up to 15 µm thick with a 0.3 µm grain size and piezoelectric module as large as $7.5 \times 10^{-12}$ C/N resulted in 35 mV of generated voltage output [49]. Another study reported the use of zinc oxide dope aluminum-based energy harvester as a cantilever structure. The effect of ZnO showed large output voltages with small displacements and was able to generate a 0.867 V$_{rms}$ output voltage [50]. This trend, however, cannot be observed in KNN and ZnO piezoelectric materials [50–52], therefore, KNN and ZnO seem more suitable for use in smaller devices.

While extensive research studies on the general functionalities of PEHs can be found in the literature, thin-film PEH devices have not been thoroughly investigated for their biocompatibility and applications in human health care systems. The present study

reports the performance comparison of the new biocompatible materials reported in the literature. The piezoelectric biocompatible materials: AlN, BaTiO$_3$, ZnO, PVDF, PZT-5H, PMN-PT and KNN-NTK were compared for the peak voltage generated, peak output power, comparable resonant frequency performance, and axial displacement performance at a volumetric body load excitation and a time-based axial displacement. The study aimed to provide a benchmark for researchers to choose the best suitable biocompatible material for thin-film piezoelectric energy harvesters based on their output voltage upon excitation. The present study will provide researchers to select more reliable, biocompatible piezoelectric materials depending on the application and the desired power output.

## 2. Materials and Methods

The current investigation aims toward analyzing a computational model to determine the electromechanical characteristics of biocompatible thin-films, namely, AlN, BaTiO$_3$, ZnO, PVDF, and KNN-NTK. Table 1 shows the mechanical and piezoelectric properties of the studied materials.

**Table 1.** The material properties of investigated biocompatible piezoelectric materials.

| Property | AlN | BaTiO$_3$ | ZnO | PVDF | PZT-5H | PMN-PT | KNN-NTK |
|---|---|---|---|---|---|---|---|
| Density (kg/m$^3$) | 3300 | 6020 | 5680 | 1780 | 7500 | 8100 | 4540 |
| Eq. Young's Modulus (GPa) | 410 | 275 | 209 | 3.8 | 127 | 112 | 8.3 |
| d31 (pC/m$^2$) | 1.91 | 34.5 | 5.4 | 30 | 274 | 760 | 10.4 |
| d33 (pC/m$^2$) | 4.95 | 85.6 | 11.6 | 25 | 593 | 1620 | 24 |
| Relative Permittivity | 9 | 1976 | 8.54 | 12.5 | 1704.4 | 1368 | 1600 |

The present investigation aimed toward analyzing a computational model to determine the electromechanical characteristics of biocompatible thin films: AlN, BaTiO$_3$, ZnO, PVDF, and KNN-NTK. The properties used in this study were derived from the COMSOL material library primarily and were cross verified with the current literature for their accuracy [51–55]. For PMN-PT (PMN–32% PT Type B), the piezoelectric properties were provided by CTS Corp The material properties of PVDF (50 nm with aluminum electrode) were obtained through the specifications as mentioned by the vendor PolyK Technologies, LLC, as the simulation model has been physically validated. With the onset of sustainable energy solutions for health monitoring devices and smart wearable sensors, it is of great performance for developers to have a scale of comparison among the popular biocompatible piezoelectric materials for easier decision-making. The literature evidently points toward the exploitation of cantilever-based energy harvesting methods, which can seldom be integrated into sophisticated and effective biomedical sensors. Such methods only explore the transverse loading conditions of the sensors and are limited to fewer models of excitation, unlike real-world applications. Therefore, in order to coalesce and supplement the development of such sensors, this study explored both the transverse vibratory and axial loading responses. A uniform thin-film strip (50 × 20 × 0.1 mm) of each sample was computationally investigated by undergoing a volumetric body load excitation and a time-based axial displacement, undergoing deformation in both the d31 and d33 mode of the piezoelectric energy generation. This allowed one to observe each mode of operation against a variety of biocompatible solutions. This was undertaken by collecting evidence in the literature of the material performance, properties, and utilizing a validated finite element multiphysics study using COMSOL 6.0.

### 2.1. Governing Equations

The electrostatics module and the circuit module work together to simulate the piezoelectric effect and the outputs. The governing equations in the coupled stress–charge form are as follows [56,57]:

$$T = c_E S - e^T E \tag{1}$$

$$D = eS + \varepsilon_s E \tag{2}$$

The vectorized form of Equations (1) and (2) are in the form:

$$
\begin{pmatrix} T_{xx} \\ T_{yy} \\ T_{zz} \\ T_{yz} \\ T_{xz} \\ T_{xy} \end{pmatrix}
= c_E \begin{pmatrix} S_{xx} \\ S_{yy} \\ S_{zz} \\ S_{yz} \\ S_{xz} \\ S_{xy} \end{pmatrix}
+ e^T \begin{pmatrix} E_x \\ E_y \\ E_z \end{pmatrix} \tag{3}
$$

$$
\begin{pmatrix} D_x \\ D_y \\ D_z \end{pmatrix}
= e \begin{pmatrix} S_{xx} \\ S_{yy} \\ S_{zz} \\ S_{yz} \\ S_{xz} \\ S_{xy} \end{pmatrix}
+ \varepsilon_s \begin{pmatrix} E_x \\ E_y \\ E_z \end{pmatrix} \tag{4}
$$

Here, '$c_E$', '$e$', and '$\varepsilon_s$' are the matrix of elasticity, coupling, and permittivity, respectively.

$$
c_E = \begin{bmatrix}
C_{11} & C_{12} & C_{12} & 0 & 0 & 0 \\
C_{21} & C_{22} & C_{23} & 0 & 0 & 0 \\
C_{31} & C_{32} & C_{33} & 0 & 0 & 0 \\
0 & 0 & 0 & C_{44} & 0 & 0 \\
0 & 0 & 0 & 0 & C_{44} & 0 \\
0 & 0 & 0 & 0 & 0 & \dfrac{C_{11} - C_{12}}{2}
\end{bmatrix}
$$

$$
e = \begin{bmatrix}
0 & 0 & e_{31} \\
0 & 0 & e_{31} \\
0 & 0 & e_{33} \\
0 & e_{15} & 0 \\
e_{15} & 0 & 0 \\
0 & 0 & 0
\end{bmatrix}^T
$$

$$
\varepsilon_s = \begin{bmatrix}
\varepsilon_{11} & 0 & 0 \\
0 & \varepsilon_{11} & 0 \\
0 & 0 & \varepsilon_{11}
\end{bmatrix}
$$

where

$$E_x = \frac{-\partial V}{\partial x}, E_y = \frac{-\partial V}{\partial y}, E_z = \frac{-\partial V}{\partial z}$$

$$S_{xx} = \frac{\partial u}{\partial x}, S_{xz} = \frac{\partial u}{\partial z} + \frac{\partial w}{\partial x}$$

$$S_{yy} = \frac{\partial v}{\partial y}, S_{yz} = \frac{\partial v}{\partial z} + \frac{\partial w}{\partial y}$$

$$S_{zz} = \frac{\partial w}{\partial z}, S_{xy} = \frac{\partial u}{\partial y} + \frac{\partial v}{\partial x}$$

where $V$ is the electrical potential, and $u$, $v$, and $w$ are the displacement field components.

*2.2. Validation*

The simulated model was first validated against the methodologies as presented by Meschino et al. and Wang et al. [58,59]. For the present simulations, the authors used a rectangular clamp-free arrangement to characterize the harvester's performance. The comparable performance data for the study of the piezo variants were obtained by utilizing the identical geometries and boundary conditions for each. With an aim toward generating a cost-effective model for further design exploration, it was found that the final mesh consisted of 10,846 elements of high quality as shown in Figure 1. To simulate biomedical stimulation with low amplitude and low frequency, transient axial-loading displacement using a piecewise cubic spline function was interpolated. The values of the applied displacements were obtained so that the stresses should not have exceeded the ultimate tensile strength or yield strength of the materials. The study ensures that each material is subjected to a safe operational cyclic loading to illustrate a practical boundary condition for real-time implementation. This study identified the peak voltage generated, peak output power, comparable resonant frequency performance, and axial displacement performance of each material.

In order to validate the model, an experiment was carried out on the PVDF piezoelectric material and the voltage output was extracted through the use of an oscilloscope. The excitation mode was time-dependent axial displacement. Figure 2 illustrates the power output versus time in which the peak voltage was roughly 11 V. The result obtained from the simulation for the same condition was 10 V, which is in good agreement with the test result.

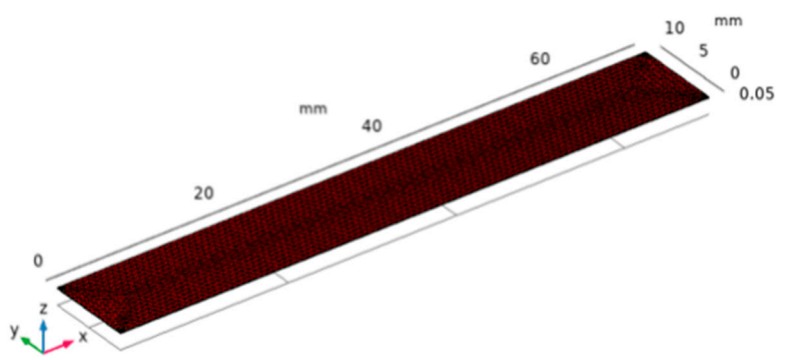

**Figure 1.** The thin film piezoelectric model (dimensions and mesh).

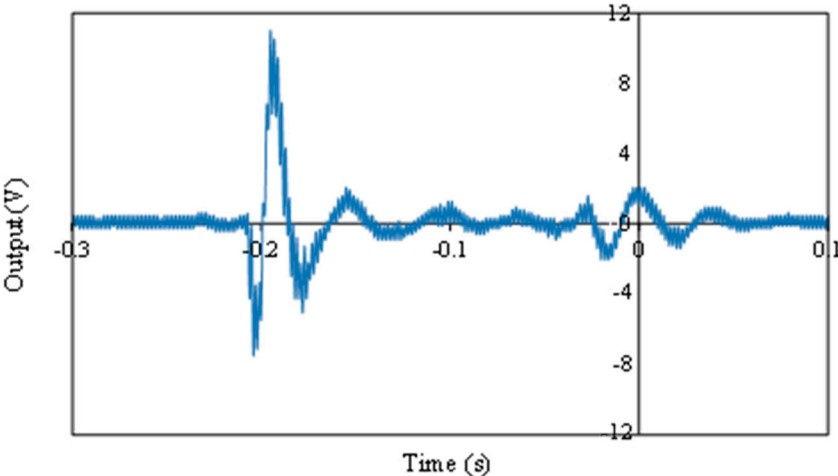

**Figure 2.** The experimentally obtained voltage versus time for PVDF undergoing axial excitation.

## 3. Results

The scope of the study was to determine the performance of five extensively employed biocompatible flexible piezoelectric materials. To put these performances in perspective, we compared their performance to popular lead-based non-biocompatible materials such as the PZT and the PMN-PT under comparable conditions. Given the results derived from this study, we observed that although the development of biocompatible materials is rapidly catching up with its predecessors, there still exists a wide chasm in their performance, which can be daunting to fill. Observing the different loading conditions, it can be inferred that there exists no best material for their robust performance in a biocompatible setting. The study identified some compelling options in the cantilever-based energy harvester configurations such as BaTiO$_3$ and PVDF in the axially-loaded configurations.

### 3.1. Transverse Vibrations

The frequency response voltage generated by the transverse vibration of a cantilever energy harvester is given in Equation (5), where $\omega_{rel}(x,t)$ is the transverse displacement relative to the clamped end of the beam [53].

$$\hat{v} = \frac{-j\omega R_l \theta_r F_r e^{j\omega t}}{\left(1 + j\omega R_l C_p^{eq}\right)(\omega_r^2 - \omega^2 + j2\zeta_r \omega_r \omega) + j\omega R_l \theta_r^2} \tag{5}$$

$$\omega_{rel}(x,t) = \frac{(1 + j\omega R_l C_p^{eq})F_r \emptyset_r(x)e^{jwt}}{\left(1 + j\omega R_l C_p^{eq}\right)(\omega_r^2 - \omega^2 + j2\zeta_r \omega_r \omega) + j\omega R_l \theta_r^2} \tag{6}$$

The power generated by the free vibration of a clamped beam without the tip mass and substrate can be calculated using Equation (7):

$$\frac{P(t)}{(-\omega^2 W_0 e^{j\omega t})^2} = \frac{1}{R_l}\left(\frac{\sum_{r=1}^{\infty}\frac{j\omega\theta_r}{\omega_r^2 - \omega^2 + j2\zeta_r \omega_r \omega}}{\frac{1}{R_l} + j\omega C_p^{eq} + \sum_{r=1}^{\infty}\frac{j\omega\theta_r^2}{\omega_r^2 - \omega^2 + j2\zeta_r \omega_r \omega}}\right)^2 \tag{7}$$

A material undergoes maximum deformation during resonance and thus is expected to produce the most piezoelectric effect. The clamp-free system is subjected to a uniform volumetric loading on one face with an acceleration of 1 g. This is intended to further improve the amplitude of deformation of a material under resonance. The obtained first natural frequencies of the materials and their respective performance measures are listed in Table 2. Torsional impacts on the model could not be identified in the mode forms until the third resonant frequency.

**Table 2.** The transverse vibration performance of the piezoelectric materials.

| Material | Resonant Frequency Hz | Peak Voltage mV | Peak Power μW |
|:---:|:---:|:---:|:---:|
| AlN | 34.359 + 0.16921i | 0.4 | $6.00 \times 10^{-8}$ |
| BaTiO$_3$ | 16.222 + 0.073209i | 2 | $1.80 \times 10^{-6}$ |
| ZnO | 16.284 + 0.078430i | 0.5 | $1.50 \times 10^{-7}$ |
| PZT-5H | 10.477 + 0.047132i | 1.6 | $1.10 \times 10^{-6}$ |
| PVDF | 4.9222 + 0.023746i | 0.1 | $5.00 \times 10^{-9}$ |
| PMN-PT | 6.2402 + 0.026850i | 2.3 | $2.60 \times 10^{-6}$ |
| KNN NTK | 2.5657 + 0.012734i | 0.9 | $4.20 \times 10^{-9}$ |

Having obtained the structure's resonant frequency, a parametric simulation with an excitation frequency ranging from 1 to 40 Hz was conducted. A voltage and power trend chart for the materials was obtained, which illustrates the harmonic excitation-based

voltage generation at various frequencies for each material in Figure 3. Due to a drastic difference in performance, the data were plotted on a log-scaled Y-axis to visualize the plot effortlessly. It was observed that PZT, PMN-PT, and BaTiO$_3$ performed remarkably under such conditions with a peak voltage of 1.6, 2.3, and 2 mV respectively. However, other materials such as AlN, ZnO, PVDF, and KNN–NTK showed poor performance when subjected to the transverse vibratory response. This was due to the inherent nature of the materials and their low d$_{33}$ attributes, as listed in Table 2. Although the performance of such materials was poor, they can still behave as high volatility active sensors with a supplementary power source in a biological environment. The natural frequency of PVDF and KNN-NTK makes it a very desirable option, making it easily susceptible to low-frequency excitation.

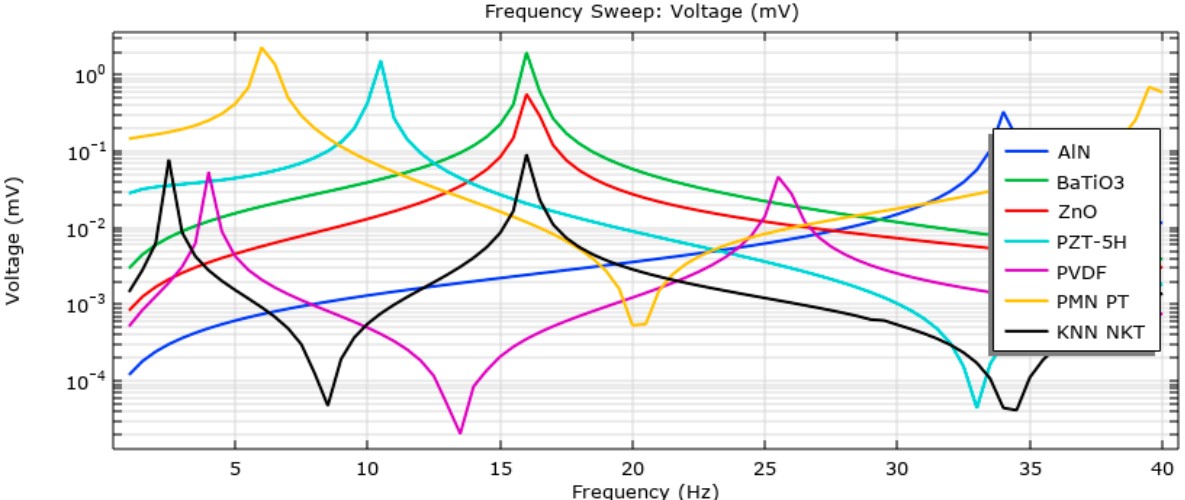

**Figure 3.** The transverse vibration response of the thin-film energy harvesters: voltage output.

A peak power output of 1.1e$^{-6}$, 2.6e$^{-6}$, and 1.8e$^{-6}$ μW was observed in PZT, PMN-PT, and BaTiO$_3$, respectively, which rendered them unusable in the form of an energy harvester at such dimensions. The output performance of these structures was plotted on a similar log-scale for comparison in Figure 4. The poor output performance of the thin-film strip was attributed to the low-charge density in the sample due to its small size. It is interesting to note that the single layer thin-films, though fascinating, failed to generate a usable amount of energy and required one to resort to a compounded structure. This method enables one to utilize such layers in a parallel- or series-configuration as required, and exploits the synergy of such structures.

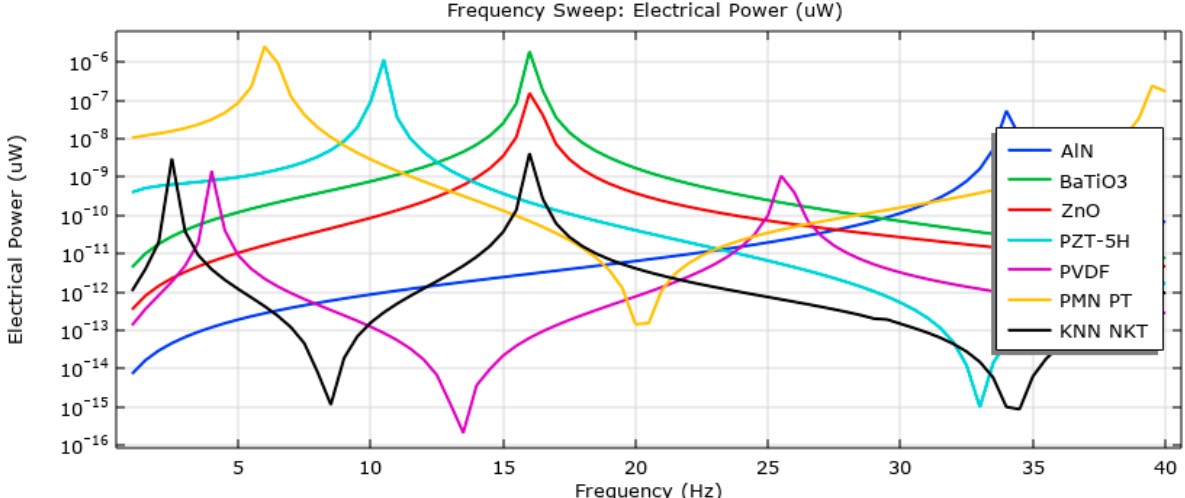

**Figure 4.** The transverse vibration response of thin film energy harvesters: power output.

The performance of thin-film energy harvesters can be further improved while utilizing various energy management methods and multilayered configurations. However, it is a challenge to subject such structures to precise excitations in real-world applications, thus it is of utmost importance to design energy harvesters that exhibit high off-resonance power generations. Based on the results of this study, it was found that most materials exhibited a bandwidth of 1 Hz. It entails that there will be a substantial drop in performance if the structure is not excited within the 1 Hz neighborhood of its resonance frequency.

We observed the conversion efficiency of the energy harvesting model while plotting the applied mechanical power to the system along with the generated power output due to the piezoelectric effect, as illustrated in Figure 5. The plots suggest that there is a substantial loss experienced during the excitation of the cantilever system when compared to the obtained electrical power.

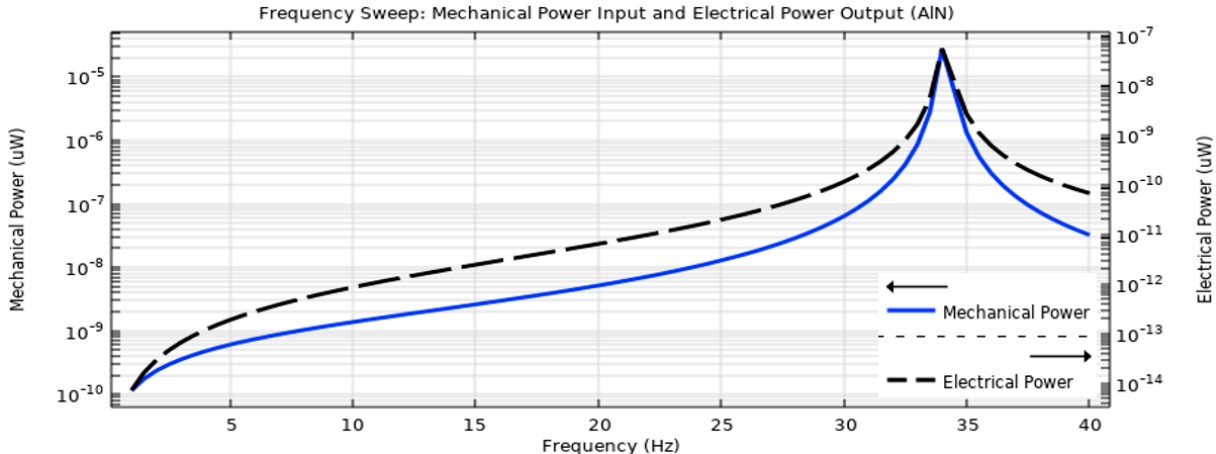

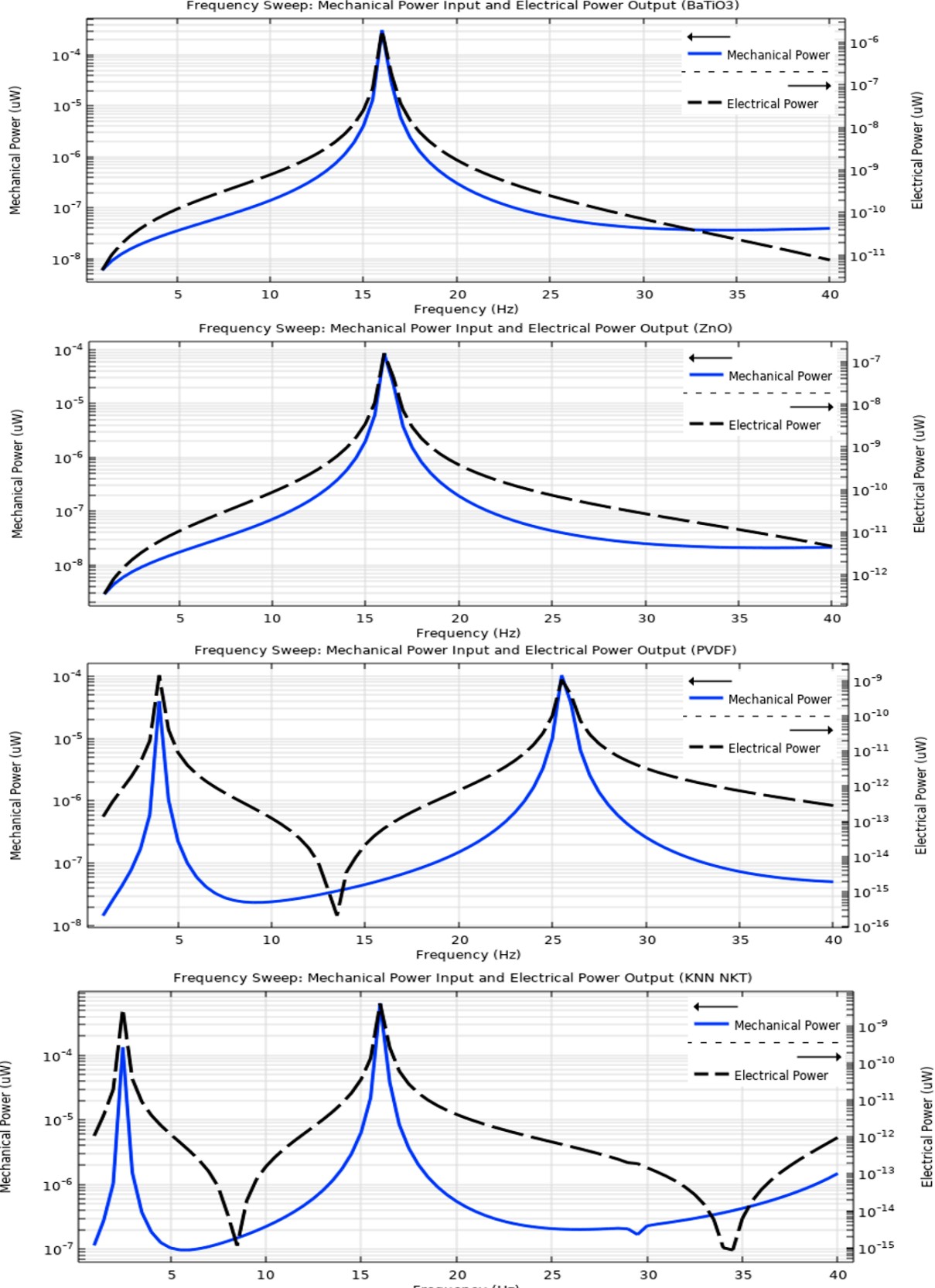

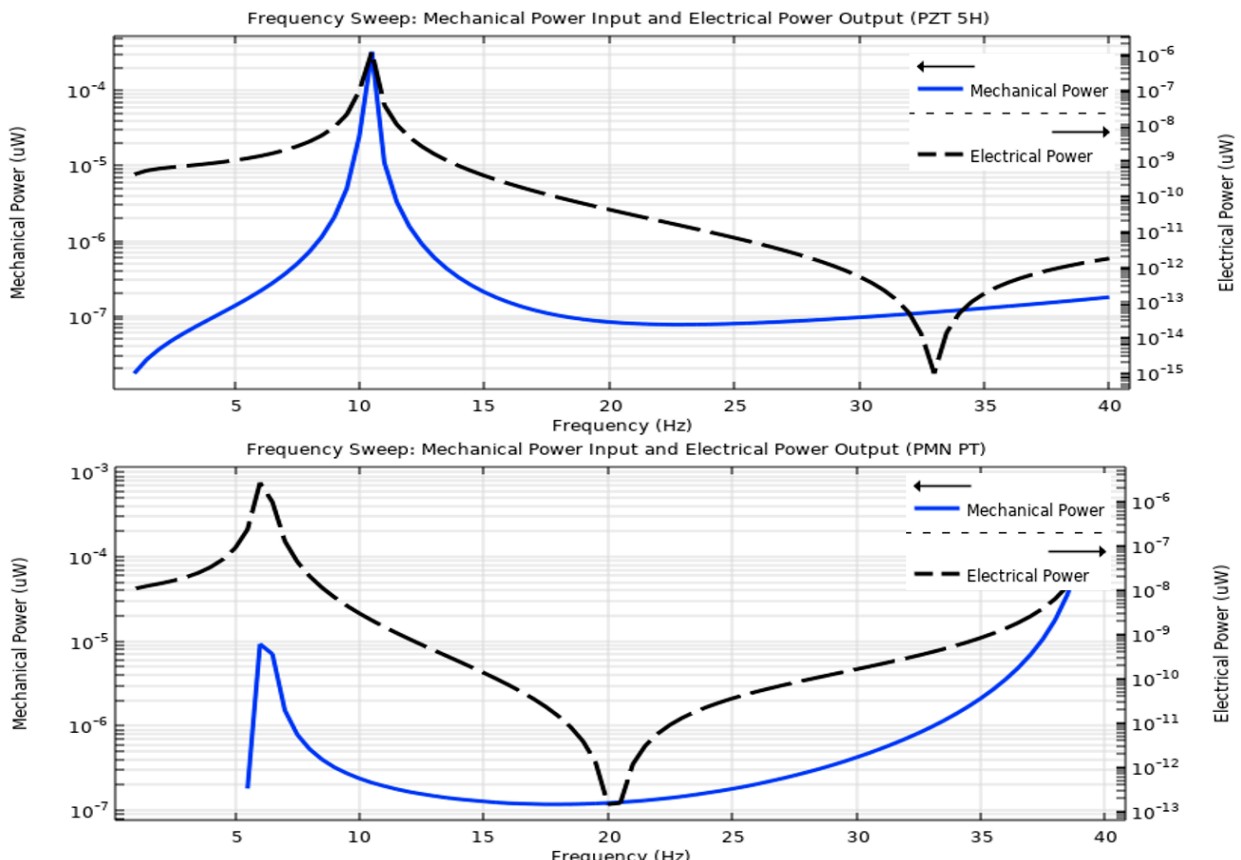

**Figure 5.** The transverse vibration mechanical power input vs. electrical power output.

### 3.2. Axial Loading

The thin-film models are excited using a time-dependent displacement function based on the calculated displacement as per the linear relationship between the stress and displacement:

$$\delta = \sigma L / E \tag{8}$$

In Equation (8), '$\delta$' is the maximum displacement value that can be applied, '$\sigma$' is the material's tensile strength, 'L' is the length, and 'E' is the Young's modulus. The values of the applied displacements were obtained so that the stresses should not have exceeded the ultimate tensile strength or yield strength of the materials. The displacements for the piezo-ceramics were found to be in the range of 0.043 to 0.18 mm, and for PVDF were 0.71 mm, as illustrated in Table 3.

**Table 3.** The axial loading performance of piezoelectric materials.

| Material | Displacement mm | Peak Voltage V | Peak Power μW |
|---|---|---|---|
| AlN | 0.053 | 0.75 | 0.28 |
| BaTiO₃ | 0.043 | 0.60 | 0.18 |
| ZnO | 0.18 | 2.60 | 3.60 |
| PZT-5H | 0.1 | 30.00 | 430.00 |
| PVDF | 0.71 | 10.00 | 50.00 |
| PMN-PT | 0.14 | 22.00 | 250.00 |
| KNN NTK | 0.07 | 0.05 | 0.001 |

The thin-film models are excited using a time-dependent displacement function using a piecewise cubic interpolation as per Equation (8and the excitation functions are plotted to illustrate the uniformity in the excitations applied to the samples in Figure A1. It is important to note that the frequency of the excitation is uniform, but varied in amplitude, so as to not exceed the tensile strength or yield strength of the materials. This is an important consideration as the FEA module continues to extrapolate the generated power output and voltage based on the calculated stress in the body. It ignores any considerations of buckling or failure of the material when subjected to high loads.

It was observed that all of the samples exhibited a higher output in comparison to when subjected to transverse vibration loading. As illustrated in Figure 6, similar to the results observed in the earlier section, we can see that very few materials had comparable performances to that of the non-biocompatible piezoceramic alternates, as shown in Table 3. Here, there was a drastic improvement in the output performance of PZT and PMN PT with an output voltage of 30 V and 22 V, respectively, with the peak electric power of 430 and 250 μW. Again, the low output power was attributed to the low charge density due to the small sample. It was also notable that the change in performance in PVDF was most drastic from 0.1 mV to 10 V. However, certain materials failed to exhibit a higher performance when compared to their counterparts, even in this mode of excitation.

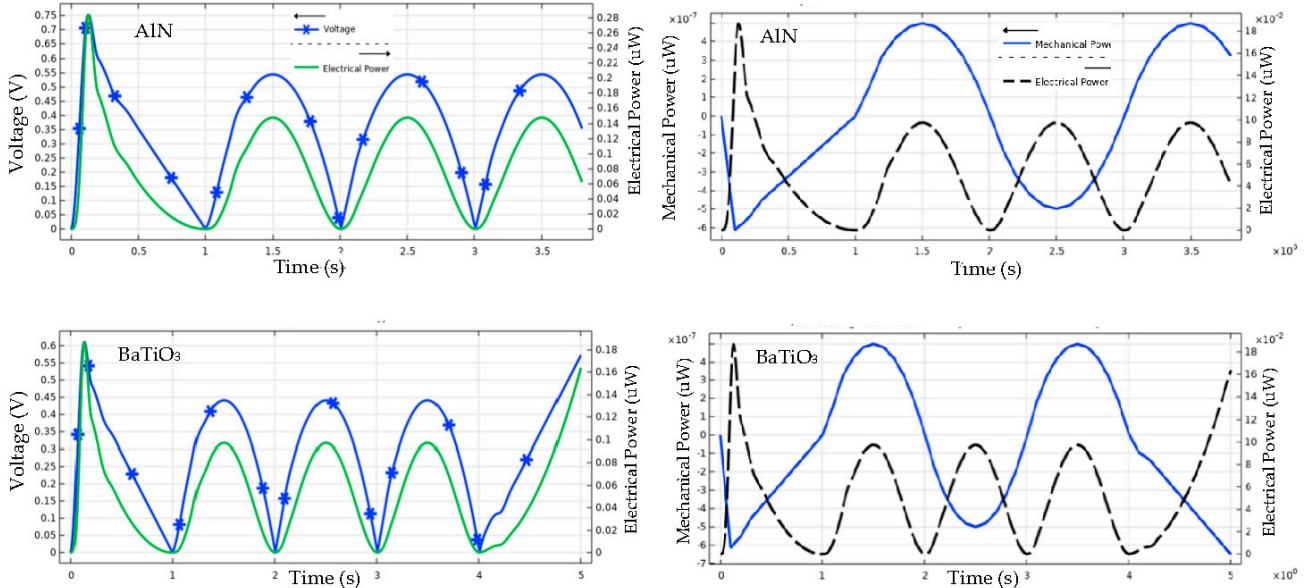

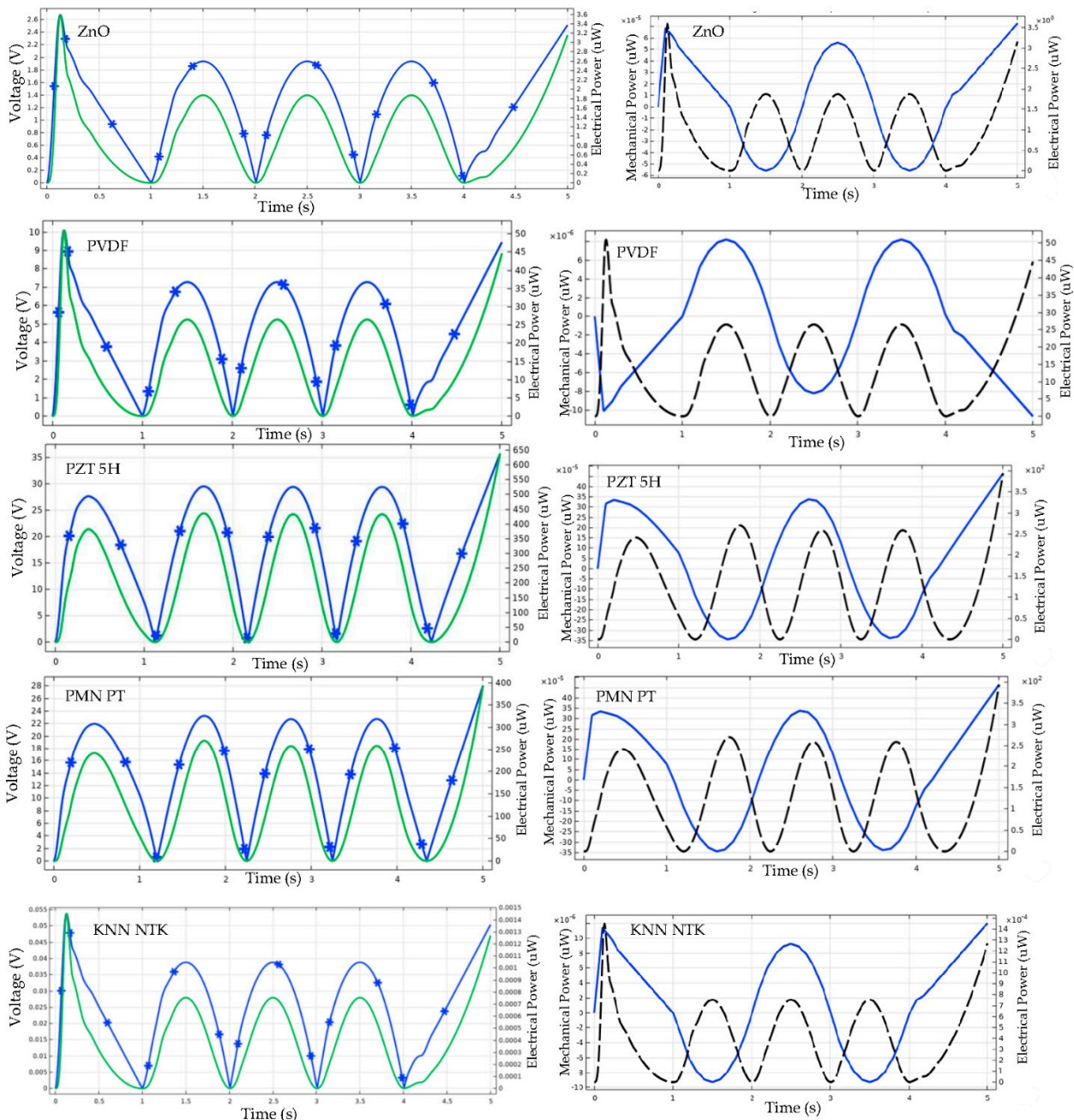

**Figure 6.** The axial-loading response of thin-film energy harvesters (**left**) and the mechanical power input vs. the electrical power output (**right**).

Similar to that of the power efficiency obtained in the previous section, we observed a phase change in the power generation mode of the energy harvesters under an axial load (Figure 6). It was observed that power was generated during both the elongation and the relaxation of the sample. Here, however, the generated power was greater than that of the applied mechanical power expended, thus illustrating the higher performance extraction of such materials.

## 4. Discussion

Piezoelectric flexible energy harvesters are extremely promising for supplying low-power mobile/portable electronics, wearable, and implantable devices as well as for autonomous and remote wireless sensors and microsystems harvesting mechanical energy from the environment and biomechanical energy from body movements. The primary

concerns of such transducers are the conversion efficiency of mechanical to electrical energy, the compactness and compliance, the cost-effectiveness, and the durability. The majority of energy harvesting devices are based on PZT thin-films; however, they require sophisticated production techniques and include a high percentage of lead (Pb), a hazardous component that limits their potential usage in flexible and wearable devices as well as medical implants. The present study analyzed the currently available variants of biocompatible, flexible, and compliant piezoelectric energy harvesters based on lead-free thin-films and compared two different loading conditions in terms of their working conditions. The energy harvester performance (produced open-circuit voltages and short-circuit currents, harvested power) of various systems is difficult to compare because of the variability of technologies and designs and the paucity of real-world analyses. Thus, the study was designed to overcome this obstacle for a comparative analysis of such materials. We found that a polymer-based piezoelectric such as PVDF performed poorly under transverse cantilever excitation with an output voltage of 0.1 V and power of $5 \times 10^{-9}$ μW, respectively. In contrast, the material outperformed its counterparts exponentially under axial-loading, while generating a voltage and power of 10 V and 50 μW, respectively. Conversely, $BaTiO_3$, being a popular piezoceramic for biomedical piezo composites, outshined in the cantilever configuration with a voltage of 2 V and power of $1.8 \times 10^{-6}$ μW, but performed poorly under axial loading with a voltage of 0.6 V and power of 0.18 μW.

Since all applied displacements were calculated based on the yield strength of the materials, they all remained in the safe region when they underwent axial displacements. PZT and PMN-PT piezo ceramics, despite offering higher power output compared with other piezoelectric materials, are not recommended in biomedical applications because they belong to the non-biocompatible material category. PVDF, on the other hand, can be considered as a suitable candidate to be used in biomedical devices due to its relatively high-power output and better stretchability.

## 5. Conclusions

Due to the many breakthroughs in material sciences, it is reasonable to infer, after a thorough examination of the data, that a sustainable energy solution is imminent. Due to their uncertain dependability and lifetime, however, it may be necessary to overcome a number of operational obstacles to order to adopt these solutions properly. Despite the development of a large variety of piezoelectric materials, the performance of popular options might still be puzzling. The study outlined some of the most essential properties of piezoelectric thin-films for early adopters, which may be easily procured and prototyped for a new generation of sustainable wearable solutions.

**Author Contributions:** Conceptualization, S.R.M., S.H.F., and T.S.; Methodology, S.R.M. and S.H.F.; Software, S.R.M.; Validation, S.R.M.; Investigation, S.R.M. and S.H.F.; Resources, K.B.; Writing—original draft preparation, S.R.M., S.H.F., and T.S.; Writing—review and editing, S.R.M. and T.S.; Visualization, S.R.M. and T.S.; Supervision, K.B.; Project administration, K.B.; Funding acquisition, K.B. All authors have read and agreed to the published version of the manuscript.

**Funding:** This research was funded by the Connaught Grant Challenge Awards for Energy Harvesting in Biomedical Applications and NSERC (ALLRP 550058–20).

**Institutional Review Board Statement:** Not applicable.

**Informed Consent Statement:** Not applicable.

**Data Availability Statement:** Not Applicable.

**Acknowledgments:** The authors would like to acknowledge the Connaught Grant Challenge Awards and NSREC for providing an opportunity to undertake this research. We would like to acknowledge the assistance of Dr. Rasool Moradi Dastjerdi and Neelesh Bhadwal for their technical insights. We would also like to thank Parham Soozandeh and Christopher Hu for their unending support in this work's undertakings.

**Conflicts of Interest:** The authors declare no conflicts of interest. The funders had no role in the design of the study; in the collection, analyses, or interpretation of data; in the writing of the manuscript, or in the decision to publish the results.

## Appendix A

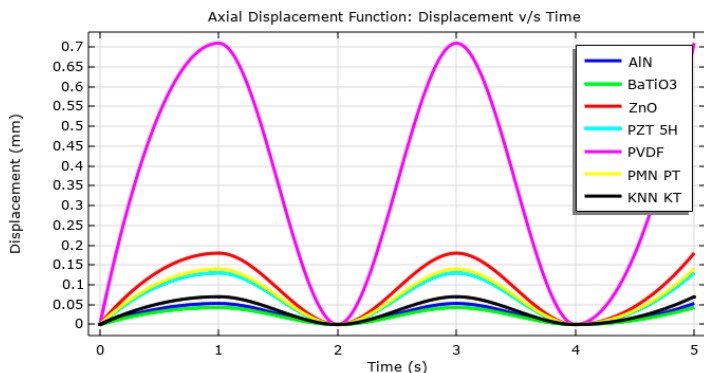

**Figure A1.** The axial-loading: displacement functions.

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
