# Peer review of "Electromechanical Performance of Biocompatible Piezoelectric Thin-Films"

_actuators, doi:10.3390/act11060171_

Round 1
Reviewer 1 Report
1. The authors report that only PVDF is used, however, PVDF is also used as a piezoelectric material and thin film with micropatterned surfaces, e.g. DOI 10.3390/POLYM11061065. The authors may also address some most recent review papers published by Pariy et. al on the application of PVDF-TrFe-based thin films.
2. I would also suggest to improve presentation of the figures, adjust font sizes for better clarity.
3. I would recommend to divide conclusion section and discussion section. In the former the most important results should be included.
4. I would also suggest to provide a clear definition a\of a thin film. In which cases a film can be considered as a thin and vice versa.
5. The durability is an important factor which affects the outcome of the experiments and energy-harvesting performance. Which materials behave better and which measures should be taken to provide better mechanical stability without any significant deterioration of the performance and service lifetime.
Author Response
Dear Reviewer,
Thank you for taking the time to review the submitted manuscript. The reviewers’ comments were very much appreciated and based on these comments, revisions were made (Highlighted Document Attached). Thank you again for your time and consideration.
The following changes have been made in line with your suggestions:
- Added appropriate descriptions and suggested citations. (Line 92 to 105)
- Figures 3,4 and 5 have been resized and replaced. However, Figure 6 despite several attempts could not be amended due to space limitations.
- Conclusion and Discussions have been divided appropriately.
- Thin FIlm definitions and considerations added. (Line 67 to 74).
- The durability and wearability of the material would require further investigative analysis under life cycle testing which we would be pursuing in our future work. However, the safe loading considerations have been highlighted for clear readability and understanding. (Lines 212, 295, 302, and 350).

Reviewer 2 Report
The paper is devoted to comparison of flexible films that made of various materials from appropriateness to energy harvesting devices. The paper is interesting and could be published but I have some minor notes:
1. One of the possible way to get energy for medical implanable devices is using biofuel cell. It will be good to add in introduction some papers. For example:
Chu T.F. et al A novel neutral non-enzimatic glucose biofuel cell based on a Pt/Au nano-alloy anode. Journal of Power Sources, 2021, v.486,229374, 10.1016/j.jpowsour.2020.229374
2. It is known the attempt to compare piezoelectric materials in term of flexoelectricity effect. It will be good to add it into introduction.
- Singhal A. et al. Comparative study of the flexoelectricity effect with a highly/weakly interface in distinct piezoelectric materials (PZT-2, PZT-4, PZT-5H, LiNbO3, BaTiO3). Waves in Random and Complex Media, 2021, 31(6), 1780-1798, 10.1080/17455030.2019.1699676
3. It will be good to add some references concerning to ZnO piezo-vibrating energy sources:
- Suhami M.I. et al. Flexible, Piezoelectric Aluminum-Doped Zinc Oxide Energy Harvesters with Printed Electrodes for Wearable Applications. International Journal of Sensors, Wireless Communications and Control, 2022, 12(1) 46-48
- Golovanov E., et al. ZnO piezoelectric films for acoustoelectronic and microenergetic applications. Coatings, 2022, v.12, 709, 10.3390/coatings12050709
- Beigh, N.T., Mallick, D. Low-Cost, High-Performance Piezoelectric Nanocomposite for Mechanical Energy Harvesting. IEEE Sensors Journal, 2021, 21(19), с. 21268-21276
4. Submit appropriate references to data from Table 1.
5. What is evidence to biocompatibility of used materials?
6. Line 226 – correct the sentence (too long)
7. It will be more appropriate combine graphs in Appendix A to one figure.
Author Response
Dear Reviewer,
Thank you for taking the time to review the submitted manuscript. The reviewers’ comments were very much appreciated and based on these comments, revisions were made (Highlighted Document Attached). Thank you again for your time and consideration.
The following changes have been made in line with your suggestions:
- Added appropriate descriptions and suggested citations.
- The data described in Table 1 is now annotated in the proceeding paragraph. (Line 173 to 178).
- Evidence of Biocompatibility is included in the form of cited work and has been added in the introduction. (Line 35)
- Line 226 is amended with 2 sentences with clear intent. (Line 279)
- Figure 1 A has been revised and combined as suggested.
